# Long-term risk of inflammatory bowel disease after endoscopic biopsy with normal mucosa: A population-based, sibling-controlled cohort study in Sweden

**Jiangwei Sun**[1]*, **Fang Fang**[2], **Ola Olén**[3,4,5], **Mingyang Song**[6,7,8], **Jonas Halfvarson**[9], **Bjorn Roelstraete**[1], **Hamed Khalili**[7,8,10], **Jonas F. Ludvigsson**[1,11,12]

1 Department of Medical Epidemiology and Biostatistics, Karolinska Institutet, Stockholm, Sweden, 2 Institute of Environmental Medicine, Karolinska Institutet, Stockholm, Sweden, 3 Clinical Epidemiology Division, Department of Medicine Solna, Karolinska Institutet, Stockholm, Sweden, 4 Sachs' Children and Youth Hospital, Stockholm South General Hospital, Stockholm, Sweden, 5 Department of Clinical Science and Education Södersjukhuset, Karolinska Institutet, Stockholm, Sweden, 6 Departments of Epidemiology and Nutrition, Harvard T.H. Chan School of Public Health, Boston, Massachusetts, United States of America, 7 Clinical and Translational Epidemiology Unit, Massachusetts General Hospital and Harvard Medical School, Boston, Massachusetts, United States of America, 8 Division of Gastroenterology, Massachusetts General Hospital and Harvard Medical School, Boston, Massachusetts, United States of America, 9 Department of Gastroenterology, Faculty of Medicine and Health, Örebro University, Örebro, Sweden, 10 Broad Institute of MIT and Harvard, Cambridge Massachusetts, United States of America, 11 Department of Pediatrics, Örebro University Hospital, Örebro, Sweden, 12 Division of Digestive and Liver Disease, Department of Medicine, Columbia University Medical Center, New York, United States of America

* jiangwei.sun@ki.se

**Data Availability Statement:** The data set cannot be shared directly under current legislation for data protection and must be requested directly from the

## Abstract

### Background

Although evidence suggests a persistently decreased risk of colorectal cancer for up to 10 years among individuals with a negative endoscopic biopsy result (i.e., normal mucosa), concerns have been raised about other long-term health outcomes among these individuals. In this study, we aimed to explore the long-term risk of inflammatory bowel disease (IBD) after an endoscopic biopsy with normal mucosa.

### Methods and findings

In the present nationwide cohort study, we identified all individuals in Sweden with a lower or upper gastrointestinal (GI) biopsy of normal mucosa during 1965 to 2016 (exposed, $n = 200,495$ and $257,192$ for lower and upper GI biopsy, respectively), their individually matched population references ($n = 989,484$ and $1,268,897$), and unexposed full siblings ($n = 221,179$ and $274,529$). Flexible parametric model estimated hazard ratio (HR) as an estimate of the association between a GI biopsy of normal mucosa and IBD as well as cumulative incidence of IBD, with 95% confidence interval (CI). The first 6 months after GI biopsy were excluded to avoid detection bias, surveillance bias, or reverse causation. During a median follow-up time of approximately 10 years, 4,853 individuals with a lower GI biopsy of normal mucosa developed IBD (2.4%) compared to 0.4% of the population references. This

respective registry holders, Statistics Sweden (information@scb.se) and the Swedish National Board of Health and Welfare (registerservice@socialstyrelsen.se), after approval by the Swedish Ethical Review Authority.

**Funding:** This study was supported by the Swedish Research Council (grant No: 2020-01706 (JFL)), FORTE (JFL), the Karolinska Institutet (FF), and the Chinese Scholarship Council (JS). The funders had no role in study design, data collection and analysis, decision to publish, or preparation of the manuscript.

**Competing interests:** I have read the journal's policy and the authors of this manuscript have the following competing interests: JFL has coordinated a study on behalf of the Swedish IBD quality register (SWIBREG) and that study received funding from Janssen corporation. OO has been PI on projects at Karolinska Institutet, financed by grants from Janssen, Takeda, AbbVie, Ferring, and Karolinska Institutet; has received fees for lectures and participation on advisory boards from Janssen, Ferring, Takeda, Bristol Myer Squibb, Galapagos, and Pfizer. OO also reports a grant from Pfizer in the context of a national safety monitoring program. JH served as speaker and/or advisory board member for AbbVie, Celgene, Celltrion, Dr Falk Pharma and the Falk Foundation, Ferring, Galapagos, Gilead, Hospira, Index Pharma, Janssen, MEDA, Medivir, MSD, Novartis, Pfizer, Prometheus Laboratories Inc., Sandoz, Shire, Takeda, Thermo Fisher Scientific, Tillotts Pharma, Vifor Pharma, UCB and received grant support from Janssen, MSD and Takeda. HK is supported by the American College of Gastroenterology Senior Research Award and the Beker Foundation; HK has received consulting fees from Abbvie and Takeda; HK has also received grant funding from Pfizer and Takeda. The other authors report no conflict of interest.

**Abbreviations:** CD, Crohn's disease; CI, confidence interval; EGD, esophagogastroduodenoscopy; ESPRESSO, Epidemiology Strengthened by histoPathology Reports in Sweden; GI, gastrointestinal; HR, hazard ratio; IBD, inflammatory bowel disease; IBD-U, IBD-unclassified; ICD, International Classification of Diseases; IR, incidence rate; PPV, positive predictive value; SNOMED, Systematized Nomenclature of Medicine; UC, ulcerative colitis.

corresponded to an incidence rate (IR) of 20.39 and 3.39 per 10,000 person-years in the respective groups or 1 extra estimated IBD case among 37 exposed individuals during the 30 years after normal GI biopsy. The exposed individuals had a persistently higher risk of overall IBD (average HR = 5.56; 95% CI: 5.28 to 5.85), ulcerative colitis (UC, average HR = 5.20; 95% CI: 4.85 to 5.59) and Crohn's disease (CD, average HR = 6.99; 95% CI: 6.38 to 7.66) than their matched population references. In the sibling comparison, average HRs were 3.27 (3.05 to 3.51) for overall IBD, 3.27 (2.96 to 3.61) for UC, and 3.77 (3.34 to 4.26) for CD. For individuals with an upper GI biopsy of normal mucosa, the average HR of CD was 2.93 (2.68 to 3.21) and 2.39 (2.10 to 2.73), compared with population references and unexposed full siblings, respectively. The increased risk of IBD persisted at least 30 years after cohort entry. Study limitations include lack of data on indications for biopsy and potential residual confounding from unmeasured risk or protective factors for IBD.

## Conclusions

Endoscopic biopsy with normal mucosa was associated with an elevated IBD incidence for at least 30 years. This may suggest a substantial symptomatic period of IBD and incomplete diagnostic examinations in patients with early IBD.

## Author summary

### Why was this study done?

- As the most frequent gastrointestinal (GI) histologic finding on endoscopy, a negative endoscopic result (i.e., normal mucosa) has been associated with a persistently decreased risk of colorectal cancer for up to 10 years. However, concerns have been raised about other long-term health outcomes among these individuals.

- We aimed to explore the long-term risk of inflammatory bowel disease (IBD) after an endoscopic biopsy with normal mucosa.

### What did the researchers find?

- In this population-based, sibling-controlled cohort study, we identified individuals with a lower ($n$ = 200,495) or upper ($n$ = 257,192) GI biopsy of normal mucosa, their individually matched general population references ($n$ = 989,484 and 1,268,897, respectively), and unexposed full siblings ($n$ = 221,179 and 274,529, respectively).

- Compared with population references and unexposed full siblings, individuals with a lower GI biopsy of normal mucosa had a persistently higher risk of overall IBD (average HR = 5.56 and 3.27, respectively), ulcerative colitis (UC, average HR = 5.20 and 3.27, respectively), and Crohn's disease (CD, average HR = 6.99 and 3.77, respectively).

- Individuals with an upper GI biopsy of normal mucosa were also at an increased risk of CD (average HR = 2.93 and 2.39, respectively).

- The elevated risk of IBD persisted at least 30 years after a biopsy with normal mucosa.

**What do these findings mean?**

- Our findings suggest a substantial symptomatic period before IBD diagnosis.

- Clinicians should be aware of the long-term increased risk of IBD in those with symptoms requiring GI investigation but with a finding of histologically normal mucosa.

## Introduction

Inflammatory bowel disease (IBD) is a chronic relapsing and remitting disease of the gastrointestinal (GI) tract [1]. Subtypes mainly encompass ulcerative colitis (UC), Crohn's disease (CD), and IBD-unclassified (IBD-U) [2]. Lower quality of life and shorter life expectancy were reported in patients with IBD than the general population likely due to digestive symptoms, extra-intestinal manifestations [3], as well as the increased risks of various malignancies [4]. Both genetic and environmental risk factors play a role in the development of IBD [5].

Previous studies have indicated that IBD may have a symptomatic period before diagnosis [6]. Already in the 1980s, Kanof and colleagues noted that adolescents with CD had impaired growth before IBD diagnosis [7]. Blackwell and colleagues demonstrated that patients with IBD often had GI symptoms 5 years before diagnosis [8]. A recent Swedish study reported that proteins related to inflammation might be up-regulated 15 years before IBD diagnosis [9]. Other dysregulated biomarkers have also been identified even 5 to 10 years prior to IBD diagnosis [10–13]. Although current guidelines stipulate that patients with suspected IBD be referred for specialist review within short time after symptom onset [14], long diagnostic delay in the symptomatic period is still a concern, prohibiting appropriate treatment [15] and leading to an increased risk of colectomy [8] and other complications [16]. Therefore, a better knowledge about this symptomatic period might not only improve disease understanding but also identify individuals who are at high risk of developing IBD.

In clinical practice, the most frequent histologic finding on endoscopy is normal mucosa. Although evidence suggests a persistently decreased risk of colorectal cancer for up to 10 years among individuals with a negative endoscopic result [17–19], an endoscopic finding of normal mucosa has been linked to excess risks of adverse health outcomes, including neurological and psychiatric disorders [20–22], chronic obstructive pulmonary disease [23], non-GI cancers [24], and mortality [25]. Therefore, concerns have been raised about the long-term health outcomes among them. Moreover, we are unaware of any studies investigating the long-term risk of IBD in those with a negative endoscopic result, which might be helpful to understand the long symptomatic period in IBD.

Taking advantage of the Swedish national healthcare registers, we conducted a cohort study, based on the nationwide histopathology cohort ESPRESSO (Epidemiology Strengthened by histoPathology Reports in Sweden) [26] and explored the risk of IBD among individuals with a GI biopsy of normal mucosa, compared with matched references randomly selected from the general population. Furthermore, because IBD as well as its protective or risk factors are likely to cluster within families, which may exert a yet unknown effect on the association between normal GI mucosa and IBD, we also compared individuals with normal mucosa to their unexposed full siblings. We hypothesized that a GI biopsy of normal mucosa was associated with an increased risk of IBD. IBD was studied as the primary outcome, while UC and CD were studied as the secondary outcomes.

## Methods

### Study design and participants

ESPRESSO contains information on all computerized GI biopsy reports from 28 pathology departments during 1965 to 2016 in Sweden, including date of biopsy, topography (upper GI tract: T60-T65 and lower GI tract: T66-T69 or T6X), and morphology (through a Swedish version of the Systematized Nomenclature of Medicine (SNOMED) coding) [26]. The exposed individuals were identified as those with a first GI biopsy report of normal mucosa (SNOMED codes: M00100 and M00110) and without other aberrations earlier (such as a biopsy record of inflammation). Our definition of normal mucosa has been validated and has a positive predictive value (PPV) of >98% [22].

In a population-matched cohort, for each exposed individual, we randomly selected up to 5 reference individuals from the Swedish Total Population Register [27], who were individually matched to the exposed individual by birth year, sex, county of residence, and calendar period. Reference individuals should be alive and biopsy-naïve at date of selection. To assess the influence of residual confounding from genetics and early environmental factors shared within families [28], we also conducted a sibling cohort to compare the risk of IBD between the exposed individuals and their unexposed full siblings. We identified the biopsy-naïve full siblings of the exposed individuals from the Swedish Multi-Generation Register (a component of the Total Population Register) [29]. The full siblings had to be alive at the biopsy date of the exposed individual. Date of biopsy for the exposed individuals and date of selection for population references or unexposed full siblings were used as the index date. Individuals with an earlier diagnosis of IBD were excluded from all analyses. The prespecified analysis plan is presented in S1 Text.

### Follow-up and ascertainment of outcome

Study participants were then linked to several Swedish national healthcare registers, using the unique personal identity number assigned to all residents in Sweden [30], with a virtually complete follow-up until an incident diagnosis of IBD, proctocolectomy (when UC was used as the outcome), emigration, death, or December 31, 2016, whichever occurred first. Some of reference individuals or biopsy-naïve full siblings were also censored when receiving a GI biopsy of normal mucosa during follow-up. Individual informed consent was waived due to the study's register-based nature [31].

We identified newly diagnosed IBD as having 1 relevant International Classification of Disease (ICD) code for IBD in the National Patient Register (available from 1964 onward) and 1 biopsy record indicating IBD in the ESPRESSO (see **Table A in S1 Appendix** for the ICD codes and histopathology SNOMED codes). A validation study has suggested a PPV of 95% (95% CI: 89% to 99%) for this diagnostic approach [32]. IBD was divided into 3 subtypes (i.e., UC, CD, or IBD-U), as outlined by Forss and colleagues [33]. As earlier research has revealed a low PPV for IBD-U [34], associations between normal GI mucosa and risk of IBD-U were not explored. For normal upper GI mucosa, only the risk of CD was assessed, since UC is limited to the rectum and colon.

### Covariates

The following covariates were considered when exploring IBD risk among individuals with a normal GI biopsy. We retrieved data on country of birth from the Total Population Register [27] (Nordic or others) and educational attainment from the Swedish Longitudinal Integrated Database for Health Insurance and Labour Market Studies [35] (available from 1990 onward;

4 groups: 0 to 9 years, 10 to 12 years, ≥13 years, and "missing," as a proxy for socioeconomic status). Number of non-primary healthcare visits, as a proxy for regular healthcare seeking behavior, was defined as the number of healthcare visits between 2 years and 6 months before the index date from the National Patient Register [36] (4 groups: 0, 1, 2 to 3, and ≥4). Healthcare visits from 6 months before until the index date was not considered to avoid overadjustment. Charlson comorbidity index [37] before the index date, without considering ulcer disease and as a proxy for general health status, was calculated according to diagnoses from the National Patient Register (3 groups: 0, 1, and ≥2). Finally, we considered history of GI diseases before the index date according to the National Patient Register (yes/no, see **Table B in S1 Appendix** for ICD codes), because endoscopy is an integral part of the management of GI diseases [38].

## Statistical analyses

Standardized difference was used to examine the balance of a covariate between the exposed and unexposed groups, and imbalance was defined as a standardized difference value greater than 0.2 [39]. We explored the risk of overall IBD, UC, and CD in individuals with a normal lower GI mucosa and risk of CD in those with a normal upper GI mucosa. Follow-up (i.e., date of cohort entry) was started 6 months after the index date to decrease potential detection bias (e.g., work up for GI biopsy increases the chance of diagnosing the other), surveillance bias (e.g., regular check-ups after a GI biopsy increase chance of early detection of IBD), and reverse causation. To estimate the average and temporal pattern of hazard ratio (HR), with 95% confidence interval (CI), comparing the exposed individuals to the matched population references and unexposed full siblings, flexible parametric survival model using the stpm2 command in Stata was applied to allow normal mucosa to vary over time (a time-varying effect) [40]. Standardized cumulative incidence of IBD was also estimated using such approach [41]. Time since date of cohort entry was used as the underlying time scale.

As the HR as well as cumulative incidence and its difference might vary with follow-up time, we presented these estimates at 6 months, 1 year, 5 years, 10 years, 20 years, and 30 years after cohort entry for each outcome. In the population-matched cohort, we conditioned the analyses on the matching variables (birth year, sex, county of residence, and calendar period) and additionally adjusted for country of birth, educational attainment, number of healthcare visits, Charlson comorbidity index, and history of GI disease. In the sibling cohort, we performed similar analyses and conditioned on family identifier as well as adjusted for birth year, sex, county of residence (collected before index date), calendar period, and the aforementioned covariables.

For normal lower GI mucosa, the risk among disease phenotypes, including CD location and UC extent, were also explored. Data on CD location and UC extent was collected at the date of IBD diagnosis and classified according to the Montreal classification [34]. CD location includes ileal (L1)/ileocolonic (L3)/unknown (LX) or colonic (L2). UC extent includes proctitis (E1)/left-sided colitis (E2), extensive colitis (E3), or extent not defined (EX) (see **Table C in S1 Appendix** for ICD codes).

## Subgroup and sensitivity analyses

To assess whether the associations would differ in different subgroup populations, we stratified the analysis by sex (male or female), age at index date (<18 y, 18 to 39.9 y, 40 to 59.9 y, and ≥60 y), and calendar period at index date (1969 to 1989, 1990 to 1999, 2000 to 2009, and 2010 to 2016). The *P* value for interaction was calculated using the Wald test for the product terms between the exposure and subgroup variables.

A number of sensitivity analyses were performed to assess the robustness of our results to potential influence from comorbidity and healthcare utilization. We restricted these analyses to: (a) individuals with a Charlson comorbidity index of zero; (b) individuals without a health-care visit between 2 years and 6 months before the index date; (c) individuals free of GI diseases before the index date; (d) individuals free of previous endoscopy (see **Table B in S1 Appendix** for relevant codes); and (e) individuals free of previous colectomy or proctocolectomy (see **Table B in S1 Appendix** for relevant codes). Given that the Prescribed Drug Register was available since July 2005, we restricted the analysis to individuals with an index date in January 2006 or later and without prior prescription of IBD medication (see **Table D in S1 Appendix** for relevant codes). To evaluate the robustness of our results to unmeasured confounding, we calculated the E-value to identify the minimum strength of the association an unmeasured confounder would need to have with both exposure and outcome to explain away the observed association [42].

Data analyses were performed using SAS version 9.4 (SAS Institute, Cary, North Carolina), Stata (version 16.1; StataCorp LP, College Station, Texas), and R version 3.6.0. A two-sided $P \leq 0.05$ was considered statistically significant. This study is reported as per the Strengthening the Reporting of Observational Studies in Epidemiology (STROBE) guideline (S1 Checklist).

### Ethics consideration

This study was approved by the Regional Ethics Review Board in Stockholm (2014/1287-31/4 and 2018/972-32).

## Results

### Lower GI biopsy and IBD

We identified 200,495 exposed individuals with a histologically normal lower GI mucosa and 989,484 matched reference individuals in the population-matched cohort, as well as 121,287 exposed individuals and their 221,179 unexposed full siblings in the sibling cohort (**Table 1**). Median age at index date was 43.1 years and 37.3 years for the exposed individuals of the 2 cohorts, respectively. Most exposed individuals were female (approximately 60%) and over two-thirds had been biopsied since the year 2000. The exposed individuals tended to have more healthcare visits, comorbidities, GI diseases, and endoscopies (e.g., colonoscopy and sigmoidoscopy) prior to the index date, as well as more colonoscopy during the follow-up than the population references and their unexposed full siblings (**Table 1**).

During a median follow-up time of approximately 10 years, 4,853 exposed individuals developed IBD (2.4%) compared to 0.4% of the population references (**Table E in S1 Appendix**). This corresponded to an incidence rate (IR) of overall IBD as 20.39 versus 3.39 per 10,000 person-years, with an IR difference of 17.01 per 10,000 person-years. The IR difference for UC and CD was 8.01 and 6.57 per 10,000 person-years, respectively (**Table E in S1 Appendix**). In the sibling cohort, the IR differences of overall IBD, UC, and CD slightly decreased to 15.87, 7.69, and 6.11 per 10,000 person-years, respectively (**Table F in S1 Appendix**).

Compared with the population references, the exposed individuals were at increased risk of IBD (overall as well as UC and CD, **Table 2 and Fig 1A–1C).** The average HR (95% CI) of overall IBD, UC, and CD was 5.56 (95% CI: 5.28 to 5.85), 5.20 (4.85 to 5.59), and 6.99 (6.38 to 7.66), respectively. The highest HR was observed shortly after cohort entry and decreased over time, but remained increased still 30 years after cohort entry (for overall IBD: HR = 2.47 (2.16 to 2.83)). The temporal pattern of HR for UC and CD was similar to that of overall IBD, although the HR was generally greater for CD than UC. For example, there was a nearly 12-fold increase in the risk of CD (HR = 11.95 (9.76 to 14.63)) at 6 months after cohort entry,

**Table 1. Characteristics of individuals with a lower GI biopsy of normal mucosa and their matched population references and unexposed full siblings.**

| Characteristics | Population matched cohort, No. (%) | | Standardized difference [f] | Sibling cohort, No. (%) | | Standardized difference [f] |
|---|---|---|---|---|---|---|
| | Normal mucosa (n = 200,495) | References (n = 989,484) | | Normal mucosa (n = 121,287) | Unexposed full siblings (n = 221,179) | |
| Age at index date, years [a] | | | | | | |
| Mean ± SD | 43.7 ± 20.4 | 43.5 ± 20.3 | 0.010 | 38.3 ± 17.6 | 39.6 ± 17.8 | −0.075 |
| Median (IQR) | 43.1 (27.4–59.5) | 42.9 (27.3–59.2) | | 37.3 (24.4–52.0) | 39.6 (25.7–53.5) | |
| <18 y | 18,132 (9.0) | 90,711 (9.2) | 0.000 | 13,442 (11.1) | 26,406 (11.9) | 0.102 |
| 18–39.9 y | 72,197 (36.0) | 357,985 (36.2) | | 52,778 (43.5) | 85,747 (38.8) | |
| 40–59.9 y | 61,392 (30.6) | 303,944 (30.7) | | 38,768 (32.0) | 76,364 (34.5) | |
| ≥60 y | 48,774 (24.3) | 236,844 (23.9) | | 16,299 (13.4) | 32,662 (14.8) | |
| Female | 121,977 (60.8) | 602,336 (60.9) | −0.001 | 73,056 (60.2) | 108,833 (49.2) | 0.223 |
| Born in Nordic country | 181,367 (90.5) | 868,132 (87.7) | −0.087 | 118,457 (97.7) | 214,564 (97.0) | −0.041 |
| Calendar period at index date [a] | | | 0.000 | | | 0.072 |
| 1969–1989 | 22,258 (11.1) | 110,151 (11.1) | | 11,514 (9.5) | 23,285 (10.5) | |
| 1990–1999 | 41,613 (20.8) | 205,595 (20.8) | | 23,807 (19.6) | 44,981 (20.3) | |
| 2000–2009 | 74,187 (37.0) | 365,619 (37.0) | | 45,257 (37.3) | 81,818 (37.0) | |
| 2010–2016 | 62,437 (31.1) | 308,119 (31.1) | | 40,709 (33.6) | 71,095 (32.1) | |
| Educational attainment | | | 0.029 | | | 0.119 |
| 0–9 y | 39,593 (19.8) | 199,031 (20.1) | | 19,525 (16.1) | 41,800 (18.9) | |
| 10–12 y | 73,745 (36.8) | 353,217 (35.7) | | 48,187 (39.7) | 85,887 (38.8) | |
| ≥13 y | 50,709 (25.3) | 248,770 (25.1) | | 32,996 (27.2) | 51,522 (23.3) | |
| Missing | 36,448 (18.2) | 188,466 (19.1) | | 20,579 (17.0) | 41,970 (19.0) | |
| History before the index date [a] | | | | | | |
| Comorbidity≥1 [b] | 46,317 (23.1) | 139,377 (14.1) | 0.233 | 23,086 (19.0) | 30,283 (13.7) | 0.145 |
| GI disease | 101,318 (50.5) | 148,984 (15.1) | 0.816 | 61,008 (50.3) | 46,516 (21.0) | 0.642 |
| Endoscopy [c] | 64,475 (32.2) | 25,433 (2.6) | 0.848 | 40,321 (33.2) | 11,687 (5.3) | 0.758 |
| EGD | 26,710 (13.3) | 23,095 (2.3) | 0.418 | 15,911 (13.1) | 9,254 (4.2) | 0.322 |
| Colonoscopy | 51,628 (25.8) | 6,111 (0.6) | 0.800 | 33,018 (27.2) | 4,205 (1.9) | 0.769 |
| Sigmoidoscopy | 7,730 (3.9) | 1,671 (0.2) | 0.265 | 4,757 (3.9) | 1,064 (0.5) | 0.236 |
| Colectomy or proctocolectomy | 221 (0.1) | 157 (0.0) | 0.038 | 134 (0.1) | 124 (0.1) | 0.019 |
| With healthcare visit [d] | 80,497 (40.2) | 260,437 (26.3) | 0.297 | 47,786 (39.4) | 62,776 (28.4) | 0.234 |
| Colonoscopy during follow-up | 20,811 (10.4) | 42,410 (4.3) | 0.235 | 12,599 (10.4) | 12,478 (5.6) | 0.175 |
| Follow-up time, years | | | | | | |
| Median (IQR) | 9.7 (4.8–17.0) | 10.1 (5.1–17.6) | −0.041 | 10.2 (5.1–17.7) | 10.8 (5.5–18.8) | −0.075 |
| 0.5–0.9 y [e] | 6,634 (3.3) | 27,009 (2.7) | 0.034 | 2,978 (2.5) | 4,107 (1.9) | 0.098 |
| 1–4.9 y | 45,185 (22.5) | 216,933 (21.9) | | 26,516 (21.9) | 44,842 (20.3) | |
| 5–9.9 y | 50,529 (25.2) | 245,337 (24.8) | | 30,257 (25.0) | 53,565 (24.2) | |
| 10–19.9 y | 61,613 (30.7) | 306,643 (31.0) | | 37,452 (30.9) | 69,289 (31.3) | |
| 20–29.9 y | 27,241 (13.6) | 144,388 (14.6) | | 17,841 (14.7) | 36,113 (16.3) | |

*(Continued)*

**Table 1.** (Continued)

| Characteristics | Population matched cohort, No. (%) | | Standardized difference [f] | Sibling cohort, No. (%) | | Standardized difference [f] |
|---|---|---|---|---|---|---|
| | Normal mucosa ($n = 200,495$) | References ($n = 989,484$) | | Normal mucosa ($n = 121,287$) | Unexposed full siblings ($n = 221,179$) | |
| ≥30 y | 9,293 (4.6) | 49,174 (5.0) | | 6,243 (5.2) | 13,263 (6.0) | |

[a] Index date: date of first biopsy record for individuals with a gastrointestinal biopsy of normal mucosa, and date of selection for their matched population references or unexposed full siblings.

[b] Measured by the Charlson comorbidity index.

[c] One individual may have multiple records of endoscopy before the index date.

[d] Defined as the number of healthcare visits between 2 years and 6 months before the index date.

[e] Follow-up started 6 months after index date.

[f] Standardized difference defined as difference in means or proportions divided by standard deviation; a covariate with a standardized difference greater than 0.2 was considered imbalanced.

EGD, esophagogastroduodenoscopy; GI, gastrointestinal; IQR, interquartile range; SD, standard deviation.

while the corresponding HR for UC was 6.40 (5.49 to 7.48) (**Table 2**). HR decreased to 3.27 (3.05 to 3.51) of overall IBD, 3.27 (2.96 to 3.61) of UC, and 3.77 (3.34 to 4.26) of CD in the sibling cohort (**Table 2**).

The cumulative incidence of IBD (overall as well as UC and CD) was higher in the exposed individuals than the population references (**Table 2** and **Table G in S1 Appendix** and **Fig 2**) and the unexposed full siblings (**Table 2** and **Table H in S1 Appendix**). The 30-year cumulative incidence of overall IBD was 3.67% (3.52% to 3.83%) in the exposed individuals and 0.96% (0.92% to 1.00%) in the matched population references (**Table G in S1 Appendix**), with a cumulative incidence difference of 2.71% (2.57% to 2.86%) (**Table 2**). Consequently, there is 1 extra estimated IBD case among 37 exposed individuals with a normal lower GI mucosa within 30 years after cohort entry. In the sibling cohort, the corresponding 30-year cumulative incidence was 4.22% and 1.56% in the exposed and unexposed siblings (**Table H in S1 Appendix**), respectively, with a cumulative incidence difference of 2.66% (**Table 2**).

In subgroup analyses by sex, age, and calendar period, the HR for overall IBD was higher in males (average HR = 6.24 (5.77 to 6.76)) than females (5.11 (4.78 to 5.46)) and in individuals aged 18 to 39.9 years (6.38 (5.91 to 6.88)) than other age groups in the population-matched cohort (both $P_{for\ interaction} < 0.001$, **Table I in S1 Appendix** for HR, **Fig A in S1 Appendix** for cumulative incidence). Compared with adults, children with a normal lower GI mucosa had a slightly different temporal pattern of HR, with a very high HR during follow-up (e.g., 15.79 (10.77 to 23.14) at 6 months and 12.49 (8.98 to 17.36) at 1 year) but a null HR at 30 years after cohort entry. This contrasted with adults who typically had a slightly lower HR (e.g., around 4 to 9 at 6 months) and then an HR around 2 to 3 at 30 years after cohort entry. In analyses of overall IBD in relation to calendar period of biopsy, the HR and cumulative incidence were lower during the latter 2 calendar periods ($P_{for\ interaction} < 0.001$), for example, within 5 years after cohort entry (**Fig 3** and **Table I in S1 Appendix** and **Fig A in S1 Appendix**), compared with earlier calendar periods. Slightly lower HRs, but with similar temporal pattern, were observed in the sibling cohort (**Table J in S1 Appendix**). Positive associations were also noted between normal lower GI mucosa and risk of different IBD phenotypes, such as colonic CD (average HR = 4.09 (3.24 to 5.17)) and UC with extensive colitis (average HR = 4.94 (4.09 to 5.96)) (**Table K in S1 Appendix**).

## Upper GI biopsy and CD

**Table L in S1 Appendix** presents the characteristics of the 257,192 individuals with a histologically normal upper GI mucosa and their 1,268,897 matched population references as well as

**Table 2. IBD during follow-up in individuals with a lower GI biopsy result of normal mucosa.**

| Outcomes | Average HR (95% CI) | Years since cohort entry | | | | | |
|---|---|---|---|---|---|---|---|
| | | 0.5 y | 1 y | 5 y | 10 y | 20 y | 30 y |
| **Compared with population references** | | | | | | | |
| Overall IBD | | | | | | | |
| Cumulative incidence difference (95% CI), % | | 0.13 (0.11–0.14) | 0.23 (0.21–0.25) | 0.82 (0.78–0.86) | 1.39 (1.33–1.45) | 2.19 (2.09–2.29) | 2.71 (2.57–2.86) |
| HR (95% CI) | 5.56 (5.28–5.85) | 8.03 (7.18–8.97) | 7.24 (6.57–7.97) | 5.29 (4.95–5.65) | 4.22 (3.97–4.49) | 3.01 (2.76–3.28) | 2.47 (2.16–2.83) |
| UC | | | | | | | |
| Cumulative incidence difference (95% CI), % | | 0.05 (0.04–0.06) | 0.10 (0.08–0.11) | 0.39 (0.36–0.42) | 0.69 (0.64–0.73) | 1.11 (1.04–1.18) | 1.32 (1.22–1.42) |
| HR (95% CI) | 5.20 (4.85–5.59) | 6.40 (5.49–7.48) | 6.06 (5.30–6.94) | 5.38 (4.91–5.88) | 4.45 (4.09–4.84) | 2.88 (2.54–3.26) | 2.15 (1.74–2.66) |
| CD | | | | | | | |
| Cumulative incidence difference (95% CI), % | | 0.06 (0.05–0.07) | 0.11 (0.09–0.12) | 0.32 (0.30–0.35) | 0.52 (0.48–0.55) | 0.76 (0.71–0.81) | 0.94 (0.86–1.02) |
| HR (95% CI) | 6.99 (6.38–7.66) | 11.95 (9.76–14.63) | 10.10 (8.48–12.04) | 6.00 (5.33–6.77) | 4.51 (4.04–5.03) | 3.53 (3.02–4.12) | 3.08 (2.47–3.85) |
| **Compared with unexposed full siblings** | | | | | | | |
| Overall IBD | | | | | | | |
| Cumulative incidence difference (95% CI), % | | 0.12 (0.10–0.14) | 0.22 (0.20–0.25) | 0.81 (0.75–0.87) | 1.37 (1.28–1.46) | 2.15 (2.01–2.29) | 2.66 (2.45–2.87) |
| HR (95% CI) | 3.27 (3.05–3.51) | 4.10 (3.51–4.79) | 3.85 (3.37–4.41) | 3.13 (2.85–3.44) | 2.77 (2.55–3.01) | 2.36 (2.09–2.67) | 2.16 (1.80–2.59) |
| UC | | | | | | | |
| Cumulative incidence difference (95% CI), % | | 0.05 (0.04–0.06) | 0.09 (0.08–0.11) | 0.37 (0.33–0.42) | 0.69 (0.63–0.76) | 1.12 (1.02–1.22) | 1.34 (1.20–1.48) |
| HR (95% CI) | 3.27 (2.96–3.61) | 3.45 (2.77–4.30) | 3.38 (2.79–4.10) | 3.42 (2.99–3.90) | 3.12 (2.78–3.50) | 2.46 (2.05–2.95) | 2.17 (1.66–2.84) |
| CD | | | | | | | |
| Cumulative incidence difference (95% CI), % | | 0.07 (0.05–0.08) | 0.11 (0.09–0.13) | 0.35 (0.31–0.38) | 0.53 (0.48–0.59) | 0.76 (0.68–0.84) | 0.94 (0.82–1.05) |
| HR (95% CI) | 3.77 (3.34–4.26) | 5.73 (4.37–7.51) | 4.88 (3.85–6.19) | 3.26 (2.76–3.84) | 2.72 (2.35–3.15) | 2.45 (1.98–3.04) | 2.31 (1.72–3.11) |

Cumulative incidence difference and hazard ratio were estimated from the flexible parametric survival model, allowing normal mucosa to vary over time. When comparing with population references, models were conditioned on matching set (birth year, sex, county of residence, and calendar period) and further adjusted for country of birth, educational attainment, number of healthcare visits, Charlson comorbidity index, and history of GI diseases. When comparing with unexposed full siblings, models were conditioned on family identifier and additionally adjusted for the abovementioned covariates.

CD, Crohn's disease; CI, confidence interval; GI, gastrointestinal; HR, hazard ratio; IBD, inflammatory bowel disease; UC, ulcerative colitis.

148,564 exposed individuals and their 274,529 unexposed siblings. During a median follow-up time of approximately 11 years, 1,048 exposed individuals developed CD (0.4%) compared to 0.1% of the population references (**Table E in S1 Appendix**). This corresponded to an IR difference of 2.39 per 10,000 person-years. In the sibling cohort, the IR difference of CD was 2.33 per 10,000 person-years (**Table F in S1 Appendix**).

Compared with the population references and unexposed full siblings, individuals with a normal upper GI mucosa had an increased risk of CD with an average HR of 2.93 (2.68 to 3.21) and 2.39 (2.10 to 2.73) (**Table 3**), respectively. The HR was highest shortly after cohort entry and remained stable 5 years after cohort entry (**Table 3 and Fig 1D**).

Higher cumulative incidence of CD was observed in the exposed individuals than the population references (**Table G in S1 Appendix and Fig 2D**) and the unexposed full siblings

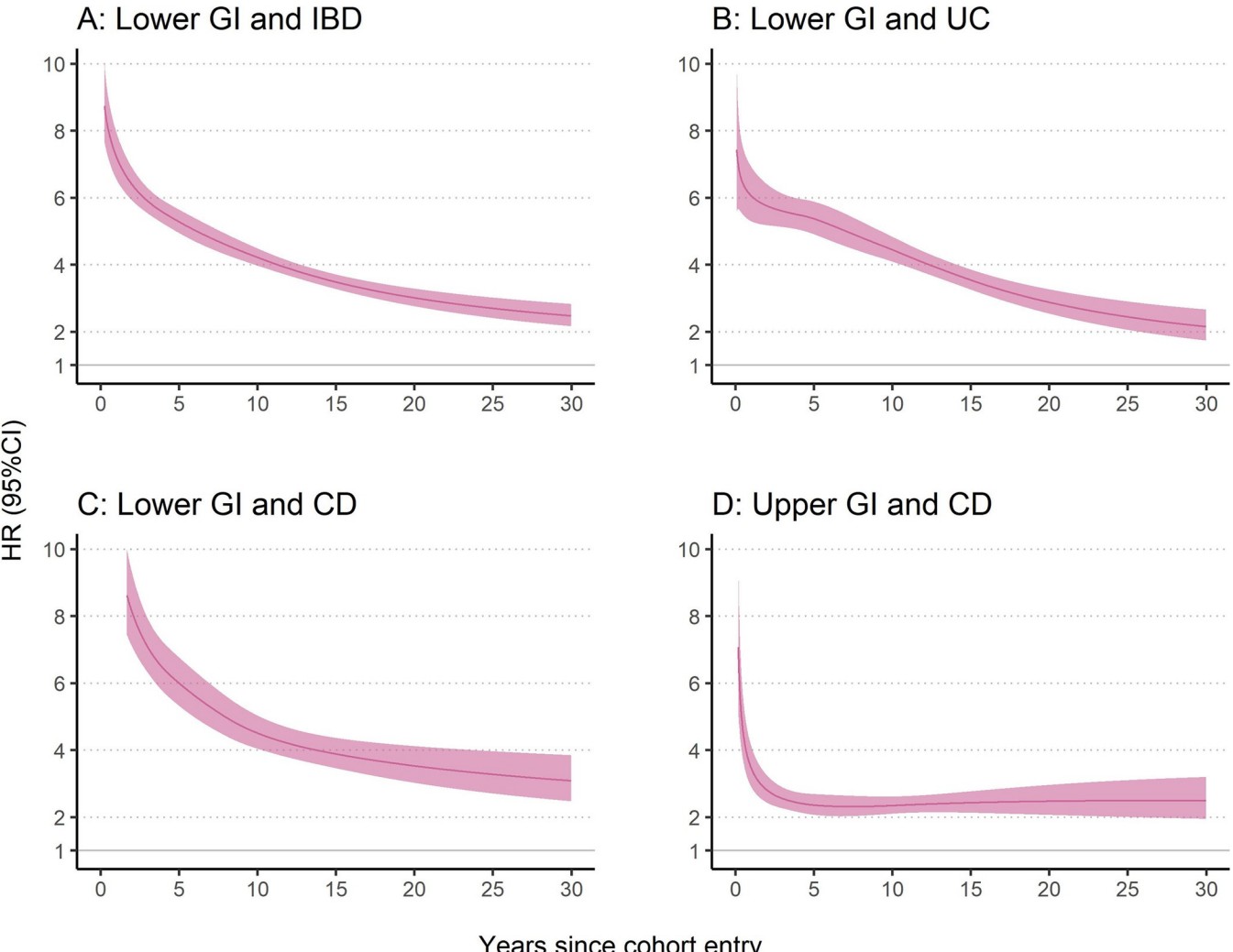

**Fig 1. HR and 95% CI of IBD as a function of time since biopsy, comparing individuals with a GI biopsy result of normal mucosa with their matched population references.** HR was estimated from the flexible parametric survival model conditioned on matching set (birth year, sex, county of residence, and calendar period) and further adjusted for country of birth, educational attainment, number of healthcare visits, Charlson comorbidity index, and history of GI diseases. Date of cohort entry was defined as 6 months after the index date. CD, Crohn's disease; CI, confidence interval; GI, gastrointestinal; HR, hazard ratio; IBD, inflammatory bowel disease; UC, ulcerative colitis.

(**Table H in S1 Appendix**). The 30-year cumulative incidence difference was 0.42% (0.35% to 0.48%) in the population-matched cohort (**Table 3**) that corresponds to 1 extra estimated CD case among 238 individuals with a normal upper GI mucosa at 30 years after cohort entry.

## Sensitivity analyses

The sensitivity analyses showed consistently higher risk of overall IBD after having a normal lower GI mucosa (**Table M in S1 Appendix**). The corresponding average HR was 5.92 (5.61 to 6.24) after restricting the analysis to individuals with a Charlson comorbidity score of zero, 6.15 (5.81 to 6.51) in individuals without an earlier healthcare visit, 5.37 (5.04 to 5.73) in those without earlier record of GI diseases, 5.65 (5.36 to 5.97) in those without earlier record of endoscopy, 5.56 (5.29 to 5.86) in those without previous colectomy or proctocolectomy, and 3.06 (2.71 to 3.45) in those with index date in January 2006 or later and without prescription of

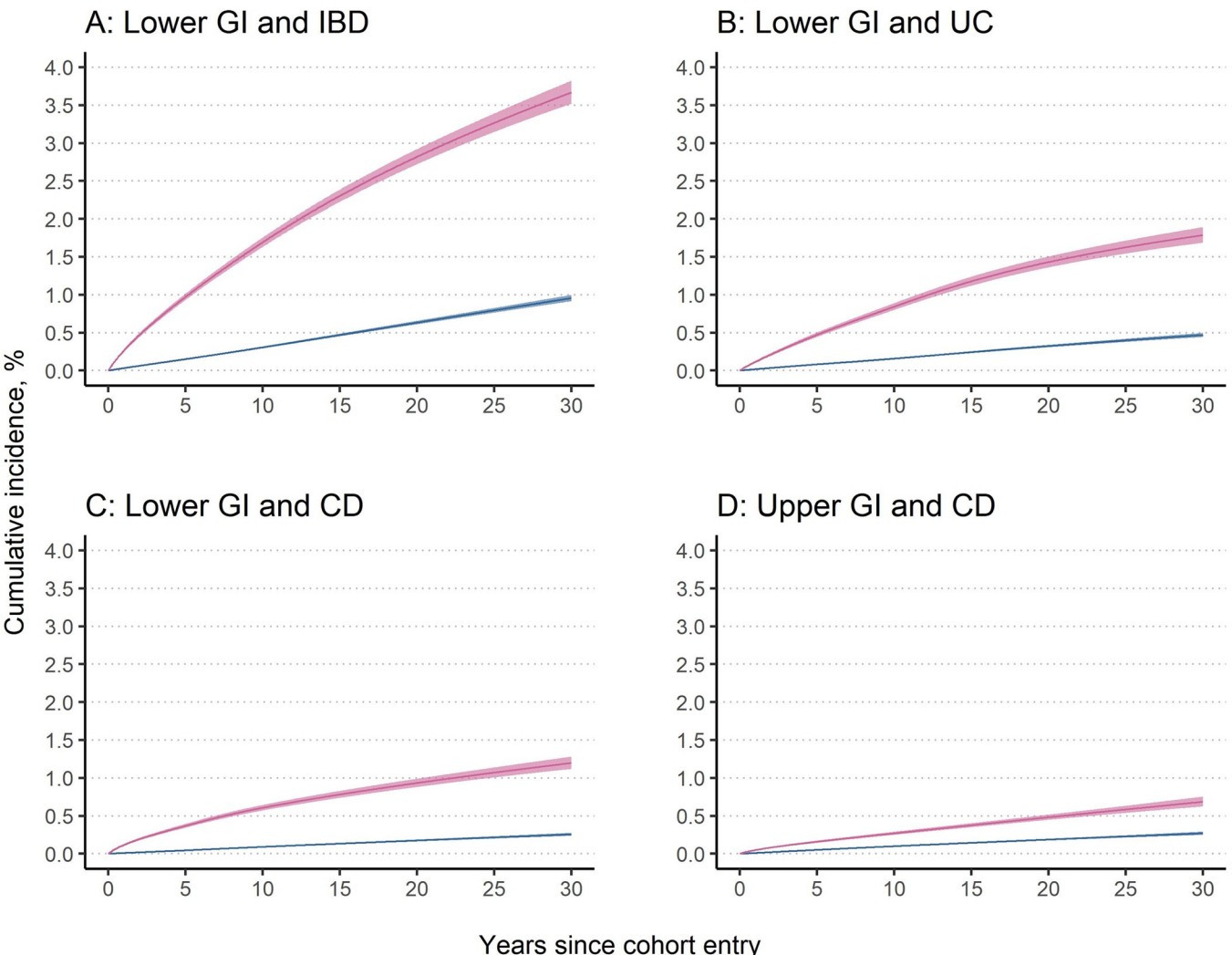

**Fig 2. Standardized cumulative incidence and 95% CI of IBD in individuals with a GI biopsy result of normal mucosa (pink) and their matched population references (blue).** The cumulative incidence was estimated from the flexible parametric survival model conditioned on matching set (birth year, sex, county of residence, and calendar period) and further adjusted for country of birth, educational attainment, number of healthcare visits, Charlson comorbidity index, and history of GI diseases. Date of cohort entry was defined as 6 months after the index date. CD, Crohn's disease; CI, confidence interval; GI, gastrointestinal; IBD, inflammatory bowel disease; UC, ulcerative colitis.

IBD medications before index date. Of note, although the HRs decreased in magnitude over time in all sensitivity analyses, all HRs remained statistically significant during the entire follow-up.

As HRs of overall IBD as well as UC and CD decreased over time, we performed a sensitivity analysis to assess the minimum degree of unmeasured confounding needed to fully explain away the associations at 30 years since cohort entry. The HR of 2.15 between normal lower GI mucosa and UC could be explained by an unmeasured confounder that confers a 3.72-fold increase in normal lower GI mucosa and a 3.72-fold risk of overall IBD. As all other HRs reported in the present study are higher than 2.15, they would therefore require a higher E-value.

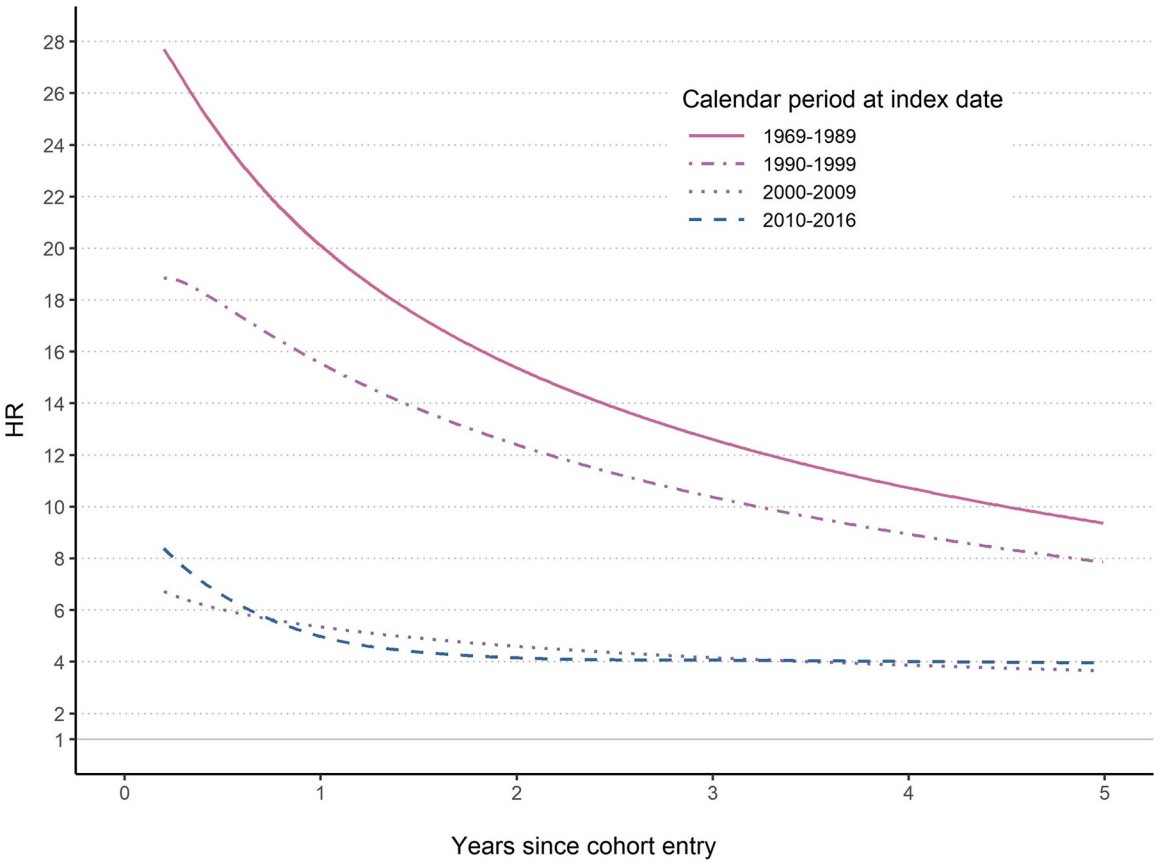

**Fig 3. HR of overall IBD as a function of time since biopsy, comparing individuals with a lower GI biopsy result of normal mucosa with their matched population references, stratified by the calendar period at index date.** Date of cohort entry was defined as 6 months after the index date. GI, gastrointestinal; HR, hazard ratio; IBD, inflammatory bowel disease.

**Table 3. CD during follow-up in individuals with an upper GI biopsy result of normal mucosa.**

| Comparison | Average HR (95% CI) | Years since cohort entry | | | | | |
|---|---|---|---|---|---|---|---|
| | | 0.5 y | 1 y | 5 y | 10 y | 20 y | 30 y |
| **Compared with population references** | | | | | | | |
| Cumulative incidence difference (95% CI), % | | 0.02 (0.02–0.03) | 0.04 (0.03–0.05) | 0.11 (0.09–0.12) | 0.17 (0.15–0.19) | 0.29 (0.26–0.33) | 0.42 (0.35–0.48) |
| HR (95% CI) | 2.93 (2.68–3.21) | 4.50 (3.67–5.51) | 3.47 (2.92–4.14) | 2.36 (2.07–2.69) | 2.34 (2.10–2.61) | 2.47 (2.07–2.96) | 2.49 (1.95–3.20) |
| **Compared with unexposed full siblings** | | | | | | | |
| Cumulative incidence difference (95% CI), % | | 0.03 (0.02–0.03) | 0.04 (0.03–0.05) | 0.12 (0.10–0.15) | 0.19 (0.15–0.22) | 0.31 (0.26–0.37) | 0.43 (0.34–0.53) |
| HR (95% CI) | 2.39 (2.10–2.73) | 3.85 (2.82–5.26) | 3.07 (2.35–4.01) | 1.90 (1.57–2.30) | 1.90 (1.62–2.23) | 2.09 (1.61–2.73) | 2.17 (1.50–3.14) |

Cumulative incidence difference and hazard ratio were estimated from the flexible parametric survival model, allowing normal mucosa to vary over time. When comparing with population references, models were conditioned on matching set (birth year, sex, county of residence, and calendar period) and further adjusted for country of birth, educational attainment, number of healthcare visits, Charlson comorbidity index, and history of GI diseases. When comparing with unexposed full siblings, models were conditioned on family identifier and additionally adjusted for the abovementioned covariates.

CD, Crohn's disease; CI, confidence interval; HR, hazard ratio.

## Discussion

In this nationwide population-based and sibling-controlled cohort study, we used 2 different measures (HR and cumulative incidence over time) to explore the temporal association between undergoing a GI diagnostic examination with normal mucosa (signaling no obvious macroscopic or microscopic aberrations) and later IBD. We found an elevated IBD incidence for at least 30 years after an endoscopic biopsy with normal mucosa. This is consistent with earlier observations of impaired growth in children and adolescents [43], up-regulated inflammatory proteins [9], dysregulated antibodies and proteins [12], and increased intestinal permeability [13] years before IBD diagnosis.

The rapid initial risk increase for IBD after cohort entry might be driven by detection bias, surveillance bias, and reverse causation (e.g., the missed IBD diagnosis at time of biopsy with normal mucosa due to incomplete endoscopies or inadequate bowel preparation). While the association between a GI biopsy with normal mucosa and IBD remained statistically significant throughout follow-up, the cumulative incidence difference increased continuously over time, leading to 1 extra estimated IBD case among 37 biopsied individuals at 30 years after cohort entry. Compared with UC (average HR = 5.20), the risk increase was greater for CD (average HR = 6.99); however, the cumulative incidence was lower for CD (1.20% versus 1.79% at 30 years after cohort entry, **Table G in S1 Appendix**). As consistent and robust results were noted in multiple subgroup and sensitivity analyses, our findings provide high-quality evidence for the long-term IBD risk of a histologically normal mucosa. Our findings may have important implications for both public health and clinical practice, as they might suggest a substantial symptomatic period before IBD diagnosis. The potential diagnostic delay during such symptomatic period might not only impair quality of life but also be the risk of complications and possibly the effectiveness of treatment [6]. Moreover, clinicians should be aware that individuals with GI symptoms requiring a GI endoscopy, although with a finding of histologically normal mucosa, are at an increased risk of future IBD, which in turn might probably help to identify individuals who are at high risk of developing IBD.

Our stratified findings indicated that the HR for overall IBD was stronger in males, in those aged 18 to 39.9 and 40 to 59.9 years, and in individuals undergoing biopsy in earlier calendar period. Of note, the magnitude of relative risk increase in IBD decreased in more recent calendar periods, probably reflecting the improved diagnosis of IBD [44,45]. For example, there has been increasing use of medical equipment with higher resolution and accuracy (e.g., capsule endoscopy, enteroscopy, and cross-sectional imaging). The endoscopic performance has also improved over time due to the training of gastroenterologist, endoscopist, or radiologist. Further, the societal awareness of IBD has also increased over time. Detection of less severe IBD in recent years, as opposed to earlier years when primarily those with severe symptoms were likely identified, has also been improved.

### Strength and limitations

Strengths of the study include the nationwide population-based and sibling-controlled cohort design with large sample size and virtually complete follow-up due to linkages to the national healthcare registers, which allowed us to present the associations up to 30 years after cohort entry and to perform a number of informative subgroup and sensitivity analyses. Both normal mucosa and IBD were objectively ascertained through validated methods with high PPVs (both ≥95%) [22,32], which greatly minimized potential information biases commonly seen in observational studies. Moreover, sibling comparisons that yielded similar results helped allay concerns about potential residual confounding from genetics and early environmental factors shared within families.

Our study also has limitations. First, lack of data on indications for biopsy is a concern. Normal mucosa might be a surrogate of a diverse range of GI symptoms or diseases before biopsy, which may be linked to IBD. Indeed, we found that individuals with normal mucosa were more likely to have a diagnosis of functional GI diseases (e.g., irritable bowel syndrome or functional dyspepsia) 5 years before biopsy in a previous study [22]. This is however unlikely to explain our results, as we noted a constantly positive association between normal mucosa and IBD during the entire 30 years of follow-up. Moreover, such concern might be partly alleviated by adjusting for Charlson comorbidity index, number of healthcare visit, and history of GI diseases in the analyses and by the similar results obtained after excluding individuals with comorbidity or GI diseases before GI biopsy. Second, incomplete coverage of inpatient care data before 1987, absence of outpatient care data before 2001, and absence of primary care information in the National Patient Register would have underestimated the number of patients with less severe IBD. Therefore, our findings should only be interpreted in the context of relatively severe IBD. On the other hand, we used all biopsies recorded by Swedish pathology departments, which should also include those referred by general practitioners. However, we cannot rule out a small number of earlier biopsies, prior to computerization of Swedish pathology departments. Third, due to the register-based nature of the study, we did not have complete information on all protective or risk factors for IBD, including lifestyle factors (e.g., smoking, body mass index, and physical activity) [5,46], medical conditions (e.g., antibiotic exposure and vitamin D deficiency) [5,32], and genetic factors (e.g., ethnicity [1,47] and risk loci (e.g., *NOD2* in European populations) [48,49]), leaving residual confounding as a concern. However, such concern might be relieved partly by the similar results from the sibling comparison. On the other hand, since genetic and early environmental factors as well as IBD are more likely to cluster within families [50], some siblings may also have undiagnosed IBD, making effect estimates in sibling comparison too conservative. Moreover, the E-value estimated in the sibling cohort after considering familial factors is also a conservative estimate and suggests that an unmeasured confounder would have to confer a 3.72-fold increase in risk of both normal mucosa and IBD to explain away the lowest HR observed at 30 years after cohort entry. Although a few microorganisms have been suggested to have such an effect on IBD (e.g., *Mycobacterium avium paratuberculosis* with CD) [5], none of the other environmental risk factors of IBD has such strong effect. Fourth, while our IBD definition has a high PPV of 95% [32], its sensitivity and specificity have not been estimated. Fifth, we lacked data on endoscopic quality, macroscopic appearance, and inflammatory markers (e.g., C-reactive protein or faecal calprotectin [51]). We therefore cannot rule out that the endoscopist might have missed a present IBD at first biopsy. However, this should not explain away our finding of a very long symptomatic period before IBD diagnosis.

In conclusion, we found that individuals with a GI biopsy of normal mucosa had an elevated risk of IBD compared with their matched population references and unexposed full siblings, with the highest risk increase noted shortly after biopsy. Although the HR decreased thereafter, it remained statistically significant throughout the 30 years of follow-up. This might suggest a substantial symptomatic period of IBD and incomplete diagnostic examinations in patients with early IBD.

## Supporting information

**S1 Checklist. STROBE Statement—checklist of items that should be included in reports of observational studies.**
(DOC)

**S1 Appendix. Supplementary figures and tables.** Fig A. Standardized cumulative incidence and 95% CI of inflammatory bowel disease in individuals with a GI biopsy result of normal mucosa (pink) and their matched population references (blue), stratified by sex, age at index date, or calendar period at index date. Table A. ICD codes and SNOMED codes defining IBD. Table B. Definitions of endoscopy, colectomy, and proctocolectomy. Table C. ICD codes assigned for phenotypes of IBD. Table D. Anatomical Therapeutic Chemical codes representing IBD treatment. Table E. Incidence rate of IBD in individuals with a GI biopsy result of normal mucosa and their matched population references. Table F. Incidence rate of IBD in individuals with a GI biopsy result of normal mucosa and their unexposed full siblings. Table G. Cumulative incidence and 95% CI of IBD during follow-up in individuals with a GI biopsy result of normal mucosa, compared with their matched population references. Table H. Cumulative incidence and 95% CI of IBD during follow-up in individuals with a GI biopsy result of normal mucosa, compared with their unexposed full siblings. Table I. Subgroup analyses of IBD during follow-up in individuals with a lower GI biopsy result of normal mucosa, compared with their matched population references. Table J. Subgroup analyses of IBD during follow-up in individuals with a lower GI biopsy result of normal mucosa, compared with their unexposed full siblings. Table K. Associations between lower GI biopsy result of normal mucosa and risk of IBD phenotypes, compared with their matched population references. Table L. Characteristics of individuals with an upper GI biopsy of normal mucosa and their matched population references and unexposed full siblings. Table M. Sensitivity analyses of IBD during follow-up in individuals with a lower GI biopsy result of normal mucosa, compared with their matched population references.
(DOCX)

**S1 Text. Analysis plan.**
(DOCX)

## Author Contributions

**Conceptualization:** Jonas F. Ludvigsson.

**Data curation:** Jonas F. Ludvigsson.

**Formal analysis:** Jiangwei Sun.

**Funding acquisition:** Jiangwei Sun, Fang Fang, Jonas F. Ludvigsson.

**Methodology:** Jiangwei Sun.

**Resources:** Bjorn Roelstraete, Jonas F. Ludvigsson.

**Software:** Jiangwei Sun.

**Supervision:** Fang Fang, Jonas F. Ludvigsson.

**Writing – original draft:** Jiangwei Sun, Jonas F. Ludvigsson.

**Writing – review & editing:** Jiangwei Sun, Fang Fang, Ola Olén, Mingyang Song, Jonas Halfvarson, Bjorn Roelstraete, Hamed Khalili, Jonas F. Ludvigsson.

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
