## [Editor Report · Decision Letter 0]

21 Sep 2022

Dear Dr Sun, 

Thank you for submitting your manuscript entitled "Long-term risk of inflammatory bowel disease after endoscopic biopsy with normal mucosa: A population-based, sibling-controlled cohort study" for consideration by PLOS Medicine.

Your manuscript has now been evaluated by the PLOS Medicine editorial staff and I am writing to let you know that we would like to send your submission out for external peer review.

Please re-submit your manuscript within two working days, i.e. by Sep 23 2022 11:59PM.

Kind regards,

Beryne Odeny

PLOS Medicine

---

## [Decision Letter · Decision Letter 1]

9 Nov 2022

Dear Dr. Sun,

Thank you very much for submitting your manuscript "Long-term risk of inflammatory bowel disease after endoscopic biopsy with normal mucosa: A population-based, sibling-controlled cohort study" (PMEDICINE-D-22-03064R1) for consideration at PLOS Medicine. 

Your paper was evaluated by a senior editor and discussed among all the editors here. It was also sent to independent reviewers, including a statistical reviewer. The reviews are appended at the bottom of this email and any accompanying reviewer attachments can be seen via the link below:

[LINK]

In light of these reviews, I am afraid that we will not be able to accept the manuscript for publication in the journal in its current form, but we would like to consider a revised version that addresses the reviewers' and editors' comments. Obviously we cannot make any decision about publication until we have seen the revised manuscript and your response, and we plan to seek re-review by one or more of the reviewers. 

We hope to receive your revised manuscript by Nov 30 2022 11:59PM. Please email us (plosmedicine@plos.org) if you have any questions or concerns.

We look forward to receiving your revised manuscript. 

Sincerely,

Callam Davidson, 

PLOS Medicine

plosmedicine.org

Please structure your abstract using the PLOS Medicine headings (Background, Methods and Findings, Conclusions).

Abstract Background: Provide the context of why the study is important. The final sentence should clearly state the study question.

In the last sentence of the Abstract Methods and Findings section, please describe the main limitation(s) of the study's methodology.

Please ensure that the study is reported according to the STROBE guideline, and include the completed STROBE checklist as Supporting Information. Please add the following statement, or similar, to the Methods: "This study is reported as per the Strengthening the Reporting of Observational Studies in Epidemiology (STROBE) guideline (S1 Checklist)."

Did your study have a prospective protocol or analysis plan? Please state this (either way) early in the Methods section.

For the adjusted analyses in Tables 2 and 3, please also provide the unadjusted analyses.

Please include definitions of the abbreviations used in Tables 2 and 3 in the legend.

In Figure 1 panels A through D, please ensure that the y axis is identical for all figures to facilitate comparison.

Comments from the reviewers:

Reviewer #1: Overview

The manuscript entitled "Long-term risk of inflammatory bowel disease after endoscopic biopsy with normal mucosa: A population-based, sibling-controlled cohort study" present interesting results describing the implication of normal mucosal biopsies and risk of IBD in a large Swedish population. Higher risk of IBD was observed in the population with a lower gastrointestinal (GI) biopsy of normal mucosa compared to a matched control population or compared to unexposed full siblings. Similar observation was obtained in individuals with an upper GI biopsy of normal mucosa. The authors suggest the existence of a substantial preclinical period of IBD and incomplete diagnostic examinations in patients with early IBD.

Introduction

-NA

Methods

-The authors mentioned that "For normal upper GI mucosa, only the risk of CD was assessed, since UC is limited to the rectum and colon." While true, the study missed the opportunity to demonstrate the negative predictive value of UC diagnosis of the International Classification of Disease coding system, which would overall reinforce the strength of this coding used in the study.

- It is unclear how the matching of the population of references was selected. A large amount of covariate are presented, and it's not clear if those covariate had equal weight in the matching process. Can you clarify?

-It seems that the author had access to the general population and not only the matched reference population. Generally speaking, a regression model applied to the entire cohort is often a more powerful tool in detecting a given effect as compared to a matched study (PMC4756459) Can the author justify the choice to select a match-control design? 

-It seems that a " flexible parametric survival model" was used, then the authors mentioned that "n the population-matched cohort, we conditioned the analyses on the matching variables (birth year, sex, county of residence, and calendar period). It's not clear what package was used or if the analysis used Cox regression or if conditional logistic regression was used to account for the matching process. Can you clarify?

-For the Sibling comparison, one of the adjusted variables was the analysis on the county of residence. Since adult's sibling may not anymore live in the same household, what was the timepoint selected for this variable?

- It seems that the authors have access to pedigree data. Was family history of IBD available in this cohort? Since healthy subject with multiple family members affected with IBD are at higher risk to develop IBD, it remains possible that the increased risk of IBD for subject with normal mucosa is confounded by being part of a multiplex.

Results

-Table 1 would benefit from a statistical comparison between the unexposed full siblings and sibling with normal mucosa. Was there an age of diagnosis difference between the unexposed full siblings and sibling with normal mucosa? 

-The comparison and results with the unexposed siblings are interesting. Did the authors compared if the age difference between sibling contribute to an increased risk of IBD? Recent publication from Spencer et al. 2020 suggests a "clustering" of IBD onset in sibling born close to each other. This study could confirm if this is the case here as well, especially given the large sample size available here.

-Was disease phenotype data available in the cohort? If yes it would be interesting to compare the different disease behavior and location.

-Was the time to disease onset similar across the unaffected sibling and the matched control population? I suspect full sibling to have more awareness of IBD and thus they might be more likely to have an earlier diagnosis than the general population.

Discussion

-When describing unmeasured risk factors, the authors mentioned that "Such concern however might be relieved partly by the similar results from the sibling comparison, as these factors are more likely to cluster within families." This not completely true for many of the described risk factor including oral contraceptive use (sibling might be of a different a different sex), or current lifestyle factors (the sibling may not be currently living in the same household).

Reviewer #2: The paper by Sun et al presents a retrospective cohort study of people who have had a GI biopsy, showing normal mucosa, matched to unexposed individuals both from the general population and from their siblings. The paper reports increased rates of IBD in the exposed group, with risk increased for as much as 30 years after the initial biopsy. This review considers the use of statistics in the paper.

The general approach of using flexible parametric survival models is appropriate, since it allows modelling of time-varying associations between exposures and outcomes, and extraction of hazard functions and ratios, and cumulative incidence, for specific subgroups. This works well for this paper. One comment: is it strictly necessary to include a six-month blanking period after the index date, if the model allows for time-varying hazards? Maybe it is - it's just a thought.

The following comments are quite minor.

The title mentions the use of sibling controls, but not the general population controls.

The methods section of the abstract talks only of lower GI biopsy, though the study does look at upper GI biopsy as well. This does not become clear until later in the paper.

The conclusion of the abstract talks of elevated risk for at least 30 years. This is conclusion is reasonable, give the results reported in the paper as a whole, but not based on the results reported in the abstract, which shows only average HRs.

In the abstract, and possibly in the paper as a whole, not much is made of the difference in results between using general population controls and sibling controls. I think it is important to note how much of the general population association can be explained by shared genetic and/or environmental factors, or whatever it is that is causing this difference.

I am sure this is the case, but I assume the Charlson scores were derived only from data available prior to the index date?

A number of subgroup analyses are performed, by stratifying the analysis. This is OK, but if possible, I prefer to see subgroup analyses in terms of a single model with an interaction between the subgroup and the exposure variables. Otherwise, the effects of other covariates can differ between the separate models, but these differences are not assessed - only the variation in association between exposure and outcome are reported. However, I accept there are different views on this topic, so it is not critical.

Would it be appropriate to cite the specific package(s) used to fit these models (if done in R)?

Reporting the number of additional cases per exposed individual is an interesting way to look at the data. Viewed in this way, how many of these could simply be thought of as false negatives amongst the original biopsies? Does this align with the reducing HRs over time, assuming that the false negative rate has improved?

Personally, I do not thing the E-value adds very much. It says how strong an association would need to be with a single covariate to explain the observed association, but in reality there may be many unknown confounders. I think most readers can interpret the magnitude of the observed association without this.

The language used throughout the paper implies that receiving a normal biopsy is a risk factor for future IBD. Whilst I am sure this is not the intention, this seems contradictory. Regardless of the causal process involved (and regardless of the E-value reported) there must be unmeasured factors that explain the observed association. For example, are the authors suggesting that if the people in the control groups of this analysis were given a biopsy (and received a normal result) at the same time as their matched exposed individual, then their risk of IBD would have been elevated? This could only be true if the act of doing a biopsy is a causal factor in the development of IBD.

On the selection of unexposed individuals, the methods section states that they were biopsy-naïve at the index date. Does this mean that a matched control could become an exposed individual at a later date? I think that would be important, to avoid an immortal time bias (if that is the correct term), and could be made more explicit if that is the case.

Reviewer #3: Thank you for inviting me to read this interesting paper on rates of IBD diagnosis following normal mucosal biopsy. It is in fact a study I would have liked to do myself if I had the time, so I was very interested to read your findings and I hope it is published after revision.

The authors used a histopathology database to identify individuals with normal colorectal mucosal biopsies and then followed them longitudinally to determine the subsequent incidence of IBD. This was 2.4% and is in itself an interesting finding. They then compared the incidence rate of IBD to a matched referent group and also a sibling cohort, and found relatively speaking that those with previous normal biopsies still had an increased risk of subsequent IBD.

Please see my comments listed below. I hope they will help improve the manuscript.

1. "Blackwell et al demonstrated that patients with IBD often had GI symptoms five years before biopsy" - I believe this study was from a large primary care dataset which did not have access to biopsy results and this should actually read "diagnosis" instead of "biopsy".

2. "an endoscopic finding of normal mucosa might still entail increased risk of adverse health outcomes, including neurological and psychiatric disorders, chronic obstructive pulmonary disease, non-GI cancers, and mortality." - "Entail increased risk" suggests having a biopsy causes that risk, rather than just being a non-causative association. This wording is too strong and needs to be changed. I assume this is only because of a bias in healthcare utilization among patients with more co-morbidites (sick patients see doctor for one problem and mention an unrelated problem which results in more healthcare utilization including colonoscopy).

This is also a potential weakness in this study - are these patients more likely be diagnosed with IBD because of a selection bias? They have already demonstrated they are more likely than average population to see a doctor? Are they by definition they are a population willing to undergo endoscopy, which in the referent group is not necessarily the case? Therefore are the results in part a reflection of increased utilization of healthcare?

3. "The exposed individuals were identified as those with a first GI biopsy report of normal mucosa" 

Is there any way of ensuring this was the patient's first ever GI biopsy, not just their first recorded biopsy in Espresso?

My understanding is that Espresso is not a comprehensive database for Sweden as it covers only 28 departments. Google tells me there are 100 hospitals in Sweden.

It is therefore very possible some of the small number of patients who later develop IBD were previously diagnosed elsewhere, were then treated, moved to a different hospital. and had a surveillance colonoscopy after with normal biopsies.cThis means it is possible these normal biopsies do not reflect pre-inflammatory IBD.

If this limitation cannot be addressed it should be listed in the limitations.

4. Do you have access to the indications for the biopsies? If so, a sensitivity analysis excluding any with "surveillance" or a previous diagnosis of IBD would be useful.

5. Do you have access to prescriptions for these patients? If so, a sensitivity analysis excluding any patients with use of any IBD drugs (mesalazine, thiopurines, systemic corticosteroids etc) before first colonoscopy would be really valuable.

6. What proportion of the referent cohort had a colonoscopy during follow-up compared with those in the previous normal biopsy group?

7. Are the code definitions of IBD validated in this dataset? Could you expand on their sensitivity and specificity. Also do these improve in more recent era, as is seen in many datasets like this - which could affect your results. What are the chances of them not picking up a diagnosis of IBD (in which case some patients may have already been diagnosed with IBD before the index biopsy and then only got coded after this)?

8. If your referent group was individually matched by birth year and calendar period then why is the median age at index date different between the cohorts? Presumably not all exposed individuals had the same number of matched referent individuals?

9. 32.2% of the exposed cohort had previously had an endoscopy. Can you specify if this was a gastroscopy, sigmoidoscopy, or colonoscopy?

Had these patients not had a normal biopsy previously if the previous procedure was not counted as the index date?

10. I note the definition of endoscopy includes colonoscopy and sigmoidoscopy - this should be split from OGD so the reader has a better understanding of the data. 

I think a sensitivity analysis, where all individuals with previous lower GI endoscopy are excluded, is necessary.

11. Why are patients with previous colectomy not excluded? Presumably the colon was removed for a reason and would not have been normal at that time so it could not be the index normal biopsy.

12. "the HR for overall IBD was higher in male (average HR=6.24 (5.77-6.76)) than female (5.11(4.78-5.46))" This comparison of Hazard Ratios is confusing to me. A ratio of ratios? I think it is better to just state the Hazard Ratios as they are without comparing them. It doesn't really mean anything to compare relative measures between them as there is no cross-over of the referent or exposed groups, they are distinct populations.

13. "in those without easier record of endoscopy" Typo - "earlier" I'm guessing?

14. "We observed that the risk increase for IBD was highest shortly after cohort entry, suggesting potential diagnostic delay." Can the authors comment on how this could have happened? Incomplete index colonoscopy but biopsies taken from normal area? Poor bowel preparation? Do they have data on these? Trainee endoscopist?

15. "Compared with UC (average HR=5.20), the risk increase was greater for CD (average HR=6.99); however, the cumulative incidence was lower for CD." This illustrates my earlier point that comparison of relative measures is not particularly helpful and the paper may represent the data better by focusing more on absolute measures like cumulative incidence. 

As a patient or clinician I am much more interested in the cumulative incidence because that tells me after a normal biopsy there is a 2.4% chance it is worth reinvestigating for IBD. The comparison to the general population is of academic interest but little clinical relevance. 

That said I found the HRs as a function of time in figure 1 interesting and useful to understand when these IBD diagnoses are more likely to occur.

16. Is "preclinical" the right word? To me preclinical implies a period where there is active disease but symptoms have not yet developed.

What the results suggest is actually the opposite - a period of being symptomatic before any histological change is detectable. I think this is really interesting but needs a different word.

Perhaps "substantial symptomatic period before IBD is detectable with current histological methods"? I am sure you will find a more eloquent way of expressing this.

[LINK]

---

## [Decision Letter · Decision Letter 2]

10 Jan 2023

Dear Dr. Sun,

Thank you very much for re-submitting your manuscript "Long-term risk of inflammatory bowel disease after endoscopic biopsy with normal mucosa: A population-based, sibling-controlled cohort study" (PMEDICINE-D-22-03064R2) for review by PLOS Medicine.

I have discussed the paper with my colleagues and the academic editor and it was also seen again by three reviewers. I am pleased to say that provided the remaining editorial and production issues are dealt with we are planning to accept the paper for publication in the journal.

[LINK]

We look forward to receiving the revised manuscript by Jan 17 2023 11:59PM.   

Sincerely,

Callam Davidson, 

Senior Editor 

PLOS Medicine

plosmedicine.org

Requests from Editors:

Please include the setting (Sweden) in the title. 

Please include headline numbers (sample size and key findings) in your Author Summary.

Please add an additional bullet under the second question in the Author Summary briefly describing the basic study design.

Please add an additional bullet under the first question in the Author Summary outlining the study question.

Please label any items cited in the Supporting Information according to our guidelines: https://journals.plos.org/plosmedicine/s/supporting-information

Comments from Reviewers:

Reviewer #1: The response to the reviewer helps to better understand the matching process, but it's still not clear to me if birth year, sex, county of residence, and calendar period had equal weight for the matching. The authors have addressed all my other concerns appropriately.

Reviewer #2: Alex McConnachie, Statistical Review

I thank the authors for their consideration of my original comments.

Regarding my comment about the need for a 6-month blanking period, the authors do not really answer this point. The HR curves over time all show a similar pattern, with highest risks early on, but a residual increased risk for up to 30 years. Would including events in the first 6 months alter this pattern at all? Or would you just see a continuation of the current trend? Perhaps all that would happen is that you would capture and measure the very high risk in those first 6 months.

That said, would the most natural explanation for the high early risk that is currently seen, be one of reverse causation? I.e. some of these people actually had disease at the initial biopsy, but this was not diagnosed. You could extend the blanking period to a year, to try to avoid this, but ultimately, where do you draw the line?

However, given the type of model used, which allows the HR to vary over time, would you end up just "chopping off" the left-hand end of the curves in Figure 1? So, in the end, it makes no difference to the HR estimates in the long term, whether or not you include events early on.

What you could say is that the HR>1 early on is more likely to be due to some sort of reverse causality or other bias, whereas the HR>1 in later periods is less likely to be affected in this way.

It is notable that the main conclusion of the paper is the IBD may have a lengthy period where patients have symptoms, before a successful diagnosis is made. In other words, the association between having a biopsy with normal mucosa and later diagnosis is due to reverse causation. I.e. patients in the early stages of the disease who have symptoms, may not be detected at biopsy, and it can take many years before a diagnosis is made.

On other matters, I still feel that the paper tends to phrase things badly, in terms of implying that having normal mucosa puts patients at increased risk. For example, line 156 says "individuals with normal mucosa had an elevated subsequent risk"; line 221 "the effect of normal mucosa"; line 256 "association between normal mucosa and IBD"; Table 2 (and Table 3) footnote "the effect of normal mucosa"; line 372-3 "higher risk of overall IBD among individuals with a normal lower GI mucosa"; line 384 "HR of 2.15 for normal lower GI mucosa"… I am sure there are other similar instances.

My issue is that it is improbable that having normal mucosa is associated with outcomes. I presume that the vast majority of the unexposed groups also had normal mucosa. The difference is that the exposed group had symptoms, sufficient to warrant undergoing a biopsy.

Otherwise, I am happy with the authors' responses to my original comments. On rereading the paper, one or two other things came up.

Table 1 now includes p-values, many of which are highly statistically significant, due to the vary large sample size. The differences between groups are not necessarily vary large (e.g. for age, comparing the exposed group with the general population, the difference is 0.2 years, but the p-value is <0.001), and may not be clinically relevant. Is there any other way to compare the groups? Some sort of standardised mean difference, perhaps?

Were the exposed group more likely to undergo biopsies after the index procedure? How many did they have? Is it possible that if you have enough biopsies, you will get a diagnosis of something eventually?

Line 385-6 says "an unmeasured confounder that confers a 3.72-fold increase in normal lower GI mucosa". Does this make sense? How can you have a confounder that "causes" normal mucosa?

Line 433-4 says "Both normal mucosa and IBD were objectively ascertained through validated methods with high PPVs (both ≥95%)." This only gives the PPV. What about the NPV? Is it known, of those with a result of normal mucosa, how many would be expected to have IBD? To what extent does this explain the observed associations?

Reviewer #3: Thank you for addressing my comments. Congratulations on a very interesting paper. I have no further comments.

[LINK]

---

## [Decision Letter · Decision Letter 3]

24 Jan 2023

Dear Dr Sun, 

On behalf of my colleagues and the Academic Editor, Professor Sanjay Basu, I am pleased to inform you that we have agreed to publish your manuscript "Long-term risk of inflammatory bowel disease after endoscopic biopsy with normal mucosa: A population-based, sibling-controlled cohort study in Sweden" (PMEDICINE-D-22-03064R3) in PLOS Medicine.

When making the formatting changes, please also address the following editorial request:

* Please include your response to Reviewer #1 ('Since the exposed individual and reference individuals were individually matched by birth year, sex, county of residence, and calendar period, these four variables had equal weight in the matching process') in the Methods.

PRESS

Sincerely, 

Callam Davidson 

Associate Editor 

PLOS Medicine